# OH spectator at IrMo intermetallic narrowing activity gap between alkaline and acidic hydrogen evolution reaction

Jiaxi Zhang [1,3], Longhai Zhang [1,3], Jiamin Liu [1], Chengzhi Zhong [1], Yuanhua Tu [1], Peng Li[2], Li Du[1], Shengli Chen [2] ✉ & Zhiming Cui [1] ✉

The sluggish kinetics of the hydrogen evolution reaction in base has resulted in large activity gap between acidic and alkaline electrolytes. Here, we present an intermetallic IrMo electrocatalyst supported on carbon nanotubes that exhibits a specific activity of 0.95 mA cm$^{-2}$ at the overpotential of 15 mV, which is 14.4 and 9.5 times of those for Ir/C and Pt/C, respectively. More importantly, its activities in base are fairly close to that in acidic electrolyte and the activity gap between acidic and alkaline media is only one fourth of that for Ir/C. DFT calculations reveal that the stably-adsorbed OH spectator at Mo site of IrMo can stabilize the water dissociation product, resulting in a thermodynamically favorable water dissociation process. Beyond offering an advanced electrocatalyst, this work provides a guidance to rationally design the desirable HER electrocatalysts for alkaline water splitting by the stably-adsorbed OH spectator.

Due to its renewable and pollution-free properties, hydrogen has been considered a promising alternative for depleted fossil fuel[1,2]. A feasible way to produce high-quality hydrogen is water electrolysis, driven by electric power from intermittent renewable energy[3,4]. Water electrolysis at low temperature can be classified into two technologies depending on the types of electrolyte: alkaline electrolyzer and proton exchange membrane electrolyzer. Although splitting water into hydrogen gas was discovered in an acidic environment, alkaline water electrolysis (AWE) represents a more mature technology and has been commercialized for more than 100 years. In particular, with the significant advances in anion exchange membrane, AWE has shown great potential for large-scale and low-cost hydrogen production[5,6]. Currently, one of the main bottlenecks for alkaline electrolyzer is the sluggish kinetics of the water-splitting reactions, especially hydrogen evolution reaction (HER) at the cathode where its kinetics is at least two orders of magnitude slower than that in acidic media[7]. In addition, the mechanism of HER in alkaline media still remains unclear and even controversial[8,9]. Apparently, it is highly desirable to fundamentally understand the mechanism and explore new, highly efficient, and stable electrocatalysts.

There are many reported factors declining the HER kinetics in base[10–12]. A key one among those is the large reaction barrier for the Volmer step $(H_2O + * + e^- \rightarrow H^* + OH^-)$ of HER involving water dissociation[13,14]. In the past decade, many efforts have been devoted to decreasing such barriers by the modification of some single-crystal electrodes[8,10,15,16]. For instance, Markovic et al. found that the decoration of $Ni(OH)_2$ clusters on Pt (111) can significantly enhance the alkaline HER activity with an improving factor of 8[17]. This was ascribed to the promotion of the water dissociation step by a bifunctional effect, where the generated OH* adsorbed at the edge of $Ni(OH)_2$ and H* adsorbed at Pt site[17,18]. Such a bifunctional effect indicates the key role of the OH binding energy (OHBE) and the integration of the bifunctional components in facilitating the HER under alkaline conditions. In the recent work of Koper's group, they found that the OHBE and HER activity at a constant hydrogen binding energy in alkaline electrolyte exhibit a volcano relationship[8], where the HER is hindered by the difficult desorption of OH at the strong OH adsorption side. Another possibility, however, should be considered that the strong OH adsorption may result in a stably adsorbed OH spectator, which does

[1]Guangdong Provincial Key Laboratory of Fuel Cell Technology, School of Chemistry and Chemical Engineering, South China University of Technology, Guangzhou 510641, China. [2]Hubei Key Laboratory of Electrochemical Power Sources, College of Chemistry and Molecular Sciences, Wuhan University, Wuhan 430072, China. [3]These authors contributed equally: Jiaxi Zhang, Longhai Zhang. ✉e-mail: slchen@whu.edu.cn; zmcui@scut.edu.cn

not directly participate in the Volmer step of HER[19]. This OH spectator may induce changes in the electronic and geometric configuration to form a new active site, which may work with another functional site via a synergistically dual-center mechanism. Ge et al. discovered that the stably adsorbed OH spectator at Fe-Co dual-atom electrocatalyst exhibited an electron-withdrawing effect for efficiently promoting the oxygen reduction reaction[20]. The interesting findings in Fe−Co dual-atom catalyst motivate us to rationally design highly efficient electrocatalyst with stably adsorbed OH spectator for the alkaline HER that previous studies have not considered. Furthermore, despite the promotions realized on the single-crystal catalysts, there are rare utilitarian electrocatalysts that can narrow the activity gap of HER between acidic and alkaline electrolytes[13,21]. One of the challenges is the precise control of the synergistically functional sites in a utilitarian electrocatalyst. In the last decade, intermetallic compounds, featuring atomically ordered structures, have emerged as a class of promising electrocatalysts, because of their remarkable performance and highly controllable compositions and surface structures[22–24]. Such unique structures endow intermetallic catalysts with integrated functional components to achieve desirable performances by the bifunctional effect or synergistically dual-center mechanism.

Herein, we present an intermetallic IrMo electrocatalyst supported on carbon nanotubes (IrMo/CNT), which performs favorable HER activity in both acid and base. Besides, the IrMo/CNT shows excellent intrinsic activity in alkaline HER with a specific activity of 0.95 mA $cm_{Ir}^{-2}$ at the overpotential of 15 mV, which is 14.4 and 9.5 times of those for commercial Ir/C (0.066 mA $cm_{Ir}^{-2}$) and Pt/C (0.1 mA $cm_{Pt}^{-2}$), respectively. More importantly, the IrMo/CNT has remarkably narrowed the HER kinetic gap between acidic and alkaline electrolytes compared with those of commercial Ir/C and Pt/C. Such a narrowed activity gap can be ascribed to the stably adsorbed OH spectator on the Mo site of IrMo, which can stabilize the water dissociation product and thereby decrease the thermodynamics barrier for alkaline HER as revealed by the density functional theory (DFT) calculations.

## Results and discussion
### Materials synthesis and characterization

The intermetallic IrMo supported on CNT was prepared by a two-step method of freeze-drying impregnation and annealing reduction, which was modified from our recently reported methods[22,23]. As can be seen in Fig. 1, the activated (hydrophilic) CNT was dispersed in a mixture solution of iridium salt and molybdate by ultrasound to form a metal salts/CNT hydrogel (Supplementary Fig. 1a). This hydrogel then went through a freeze-drying treatment to remove the water and form metal salts/CNT aerogel (Supplementary Fig. 1b). Such aerogel was finally annealed under an $H_2$/Ar flow to obtain the final intermetallic IrMo/CNT electrocatalyst. The CNT-supported intermetallic IrMo presents a structure type of $B19$-AuCd and a space group of $Pmma$[25]. The

diffraction peaks around 37.7°, 40.8°, and 42.9° can be indexed to (002), (200), and (111) facets of the intermetallic IrMo (see Fig. 2a). The crystal structure of the intermetallic IrMo is quite different from the metallic Ir, which presents a space group of $Fm\text{-}3m$ (Fig. 2a)[14]. The valance band spectra of IrMo/CNT show a positive shift relative to Ir/C (Fig. 2b). This indicates that the valance band energy of IrMo shifts away from Fermi level compared with that of Ir[4,26]. The Ir $4f$ XPS of IrMo shows more $Ir^0$ species than that of Ir (Fig. 2c). This may have resulted from the transfer of electrons from Mo to Ir in IrMo, which would lead to the negative shift of the $d$-band center of Ir. High angle annular dark field/scanning transmission electron microscope (HAADF/STEM) was employed to evaluate the morphology and crystal structure of IrMo/CNT (Fig. 2d–f). The prepared IrMo intermetallic nanoparticles (NPs) are well dispersed on CNT support (Fig. 2d) and mostly size between 1.5 and 2.5 nm. The average size of the NPs is 2.1 nm. Figure 2e illustrates the atomic-resolution HAADF/STEM images of the intermetallic IrMo NPs. The lattice spacing of 0.220, 0.240, and 0.213 nm in these images can be indexed to the (200), (002), and (111) planes of intermetallic IrMo. This is consistent with the fast Fourier transform diffraction results of the related IrMo NPs. The atomic-level insight of the IrMo NPs indicates that three kinds of exposed facets are detected at the surface, including (001), (100), and (111) facets. The line scan of an IrMo NPs shows that the Ir and Mo are uniformly distributed in the particles (Fig. 2f). This indicates that there is no segregation of Ir or Mo shell at the surface.

### HER measurement

The HER activity in the base of IrMo/CNT, Ir/C, and Pt/C electrocatalysts was first measured in $H_2$-satruated 1 M KOH at a rotating disk electrode under 1600 rpm. As shown in Fig. 3a, the IrMo/CNT shows the lowest HER onset potential among the three evaluated catalysts. The IrMo/CNT merely needs an overpotential ($\eta_{10}$) of 17 mV to achieve the current density of 10 mA $cm_{geo}^{-2}$, which is much lower than those of Pt/C (46 mV) and Ir/C (70 mV) (Fig. 3e). Tafel plots derived from the linear scanning voltammetry (LSV) curves reveal that the IrMo/CNT shows a much smaller Tafel slope (22 mV $dec^{-1}$) compared with those of Pt/C (99 mV $dec^{-1}$) and Ir/C (122 mV $dec^{-1}$) (Fig. 3f and Supplementary Fig. 2), indicating the significantly enhanced alkaline HER kinetics of IrMo/CNT. Besides, the IrMo/CNT exhibits a superior alkaline HER activity (the smaller $\eta_{10}$ and Tafel slope) relative to the typical HER electrocatalysts reported in literature (Fig. 3b and Supplementary Table 1). Considering the different loading of the catalysts on electrodes, the comparison with the overpotential at a geometric current density or the Tafel slope is inaccurate and questionable. A better descriptor for HER activity could be the turnover frequency (TOF) per active site. Thus, we have calculated the TOF of the evaluated electrocatalysts according to the reported method[27,28]. As revealed by the comparison between the calculated TOF of IrMo/CNT and those of the reported electrocatalysts, the IrMo/CNT exhibits a superior HER

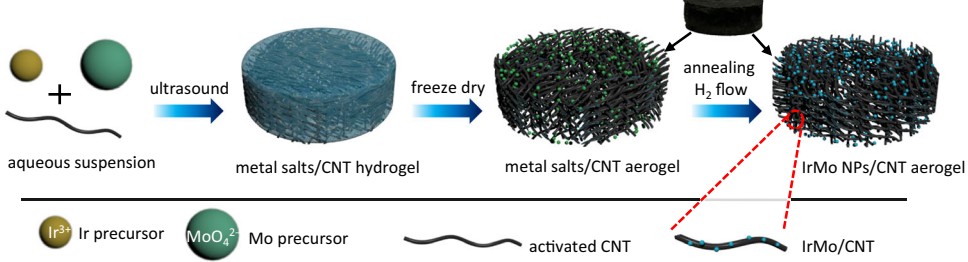

**Fig. 1 | Schematic of the preparation for the CNT-supported Intermetallic IrMo NPs by freeze-drying impregnation and annealing reduction.** Hydrophilic groups at the activated CNT can help the formation of hydrogel during the ultrasound treatment by the crosslinking of CNT. The water in the frozen hydrogel complexes was sublimated during the freeze drying and the crosslinking CNT framework was preserved to form the aerogel. In the meanwhile, the metal precursors were uniformly dispersed on the support. Such aerogel could preserve the shape even after being annealed under reductive atmosphere.

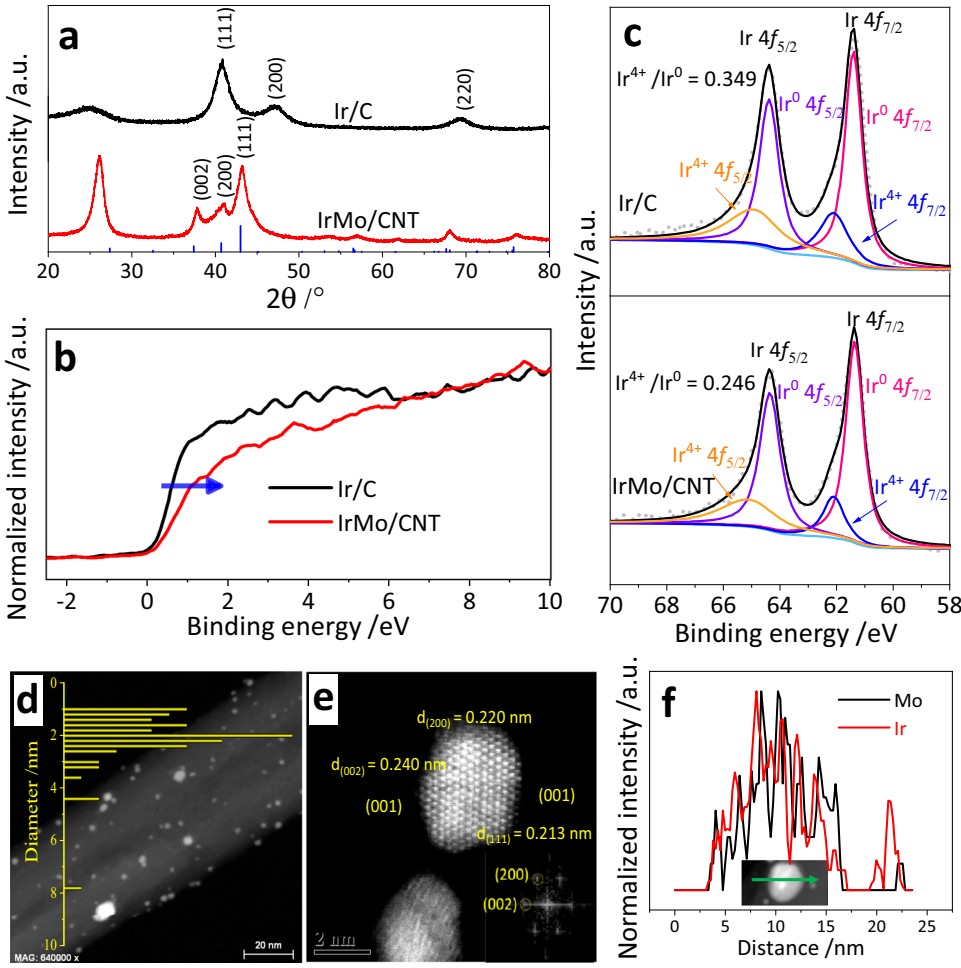

**Fig. 2 | Physical characterization. a** XRD patterns. The blue vertical line shows the PDF of IrMo[25]. **b** Valance band spectra. **c** Ir 4f XPS spectra. **d** HAADF/STEM image. The inserted bar graph shows the NPs size distribution for IrMo. **e** Atomic-resolution HADDF/STEM image of intermetallic IrMo nanoparticles, the inserted images show the fast Fourier transform (FFT) diffraction of IrMo NPs. **f** HADDF/STEM line scan of an intermetallic IrMo nanoparticle.

rate (15.4 $H_2$ $s^{-1}$ per active site) in base among the benchmarking electrocatalysts (Fig. 3c).

The HER measurement in acidic condition was performed in $H_2$-saturated 0.5 M $H_2SO_4$. As shown in Fig. 3d, the IrMo/CNT also presents the lowest onset HER potential among the three measured electrocatalysts as it does in alkaline condition. The Tafel slope of IrMo/CNT in acidic HER is comparable to Pt/C and Ir/C, indicating their similar HER kinetics in acid condition (Supplementary Fig. 3). Furthermore, the IrMo/CNT shows a highly durable HER performance both in alkaline and acid electrolytes (see Supplementary Figs. 4–12 and Supplementary Note 1). It is worth noting that the HER rate and kinetics of the commercial electrocatalysts (Pt/C and Ir/C) exhibit a distinct gap between acidic and alkaline electrolytes as revealed by the measured different $\eta_{10}$ and the Tafel slope (Fig. 3e, f). Interestingly, such activity and kinetics gap are remarkably narrowed by IrMo/CNT. A further analysis of the narrowed kinetics gap will be presented below.

### Intrinsic activity gap between alkaline and acidic HER

The specific activities of the evaluated catalysts were obtained via normalizing the HER current by their electrochemical surface areas (ECSA) that are 41, 62, and 68 $m^2$ $g_{Ir\ or\ Pt}^{-1}$ for IrMo/CNT, Ir/C, and Pt/C (Supplementary Fig. 13), respectively. The normalized specific current density as a function of the potential in acidic and alkaline HER is plotted in Fig. 4a. According to the overall comparison of the specific activity, it can be seen clearly that the IrMo/CNT shows a quite different activity gap between acidic and alkaline HER compared with

those of Pt/C and Ir/C. The HER-specific current densities of Ir/C and Pt/C in the base are much lower than those in acid under the same potential. Impressively, such sharp deterioration of the intrinsic activity is not found for IrMo/CNT.

For a quantitative comparison, the specific current densities at $\eta = 15$ mV were extracted and plotted in Supplementary Fig. 14. The IrMo/CNT shows the highest specific activity in both acidic (2 mA $cm_{Ir}^{-2}$) and alkaline (0.95 mA $cm_{Ir}^{-2}$) electrolytes, which are 3.6 and 14.4 folds of those for Ir/C, 4.6 and 9.5 folds for Pt/C, respectively. TOF is a convincing descriptor for intrinsically comparing the activity per active site of catalysts and being widely used for estimating the activity of electrocatalysts in recent years[29–31]. The calculated TOF as a function of potential for the evaluated electrocatalysts is plotted in Supplementary Figs. 15–17. The TOFs at $\eta = 15$ mV are extracted and compared in Fig. 4b. The IrMo/CNT in acidic HER shows the highest TOF of 4.8 $H_2$ $s^{-1}$ per active site among all the evaluated performances. What impresses us is that the HER TOF of IrMo/CNT in the base at $\eta = 15$ mV also reaches at a high value of 2.3 $H_2$ $s^{-1}$ per active site, which is 14.4 and 10 folds of those for Ir/C and Pt/C, respectively. Such a high TOF of IrMo/CNT in alkaline media also surpasses the HER TOF of Ir/C and Pt/C in acid electrolyte.

The intrinsic Tafel plots derived from the current density normalized by ECSA for the evaluated electrocatalysts in acid and base are further compared in Supplementary Figs. 18–20. It can be found that the IrMo/CNT intrinsically shows the lowest Tafel slope in alkaline HER among the evaluated electrocatalysts and a comparable Tafel slope

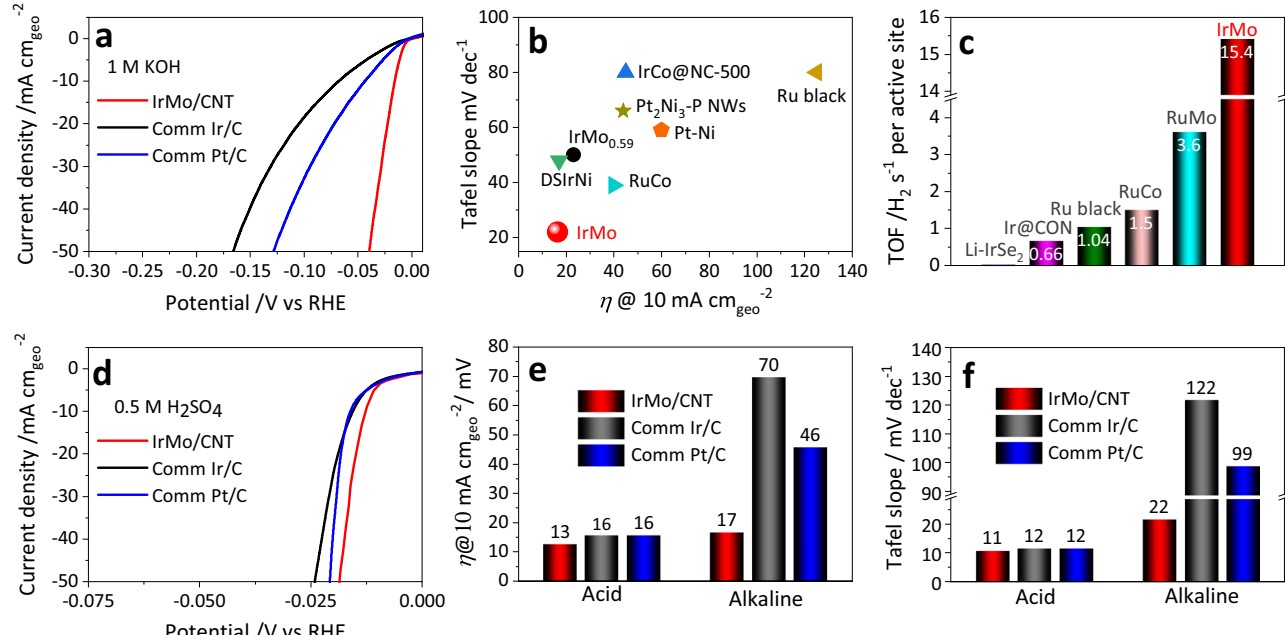

**Fig. 3 | HER performance of IrMo/CNT, commercial Ir/C, and Pt/C. a** HER polarization curves in 1 M KOH. **b** Comparison of the HER Tafel slope and the overpotential ($\eta$) at 10 mA cm$^{-2}$ ($\eta_{10}$) in 1 M KOH of IrMo/CNT with some reported precious electrocatalysts[14,45–50]. **c** Comparison of the HER TOF of IrMo/CNT with the reported efficient HER electrocatalysts at $\eta = 50$ mV in alkaline condition[3,47,48,51,52]. **d** HER polarization curves in 0.5 M H$_2$SO$_4$. **e** Comparison of the extracted $\eta_{10}$ from **a**, **d** between acidic and alkaline electrolytes. **f** Comparison of the Tafel slope derived from **a**, **d** between acidic and alkaline electrolytes.

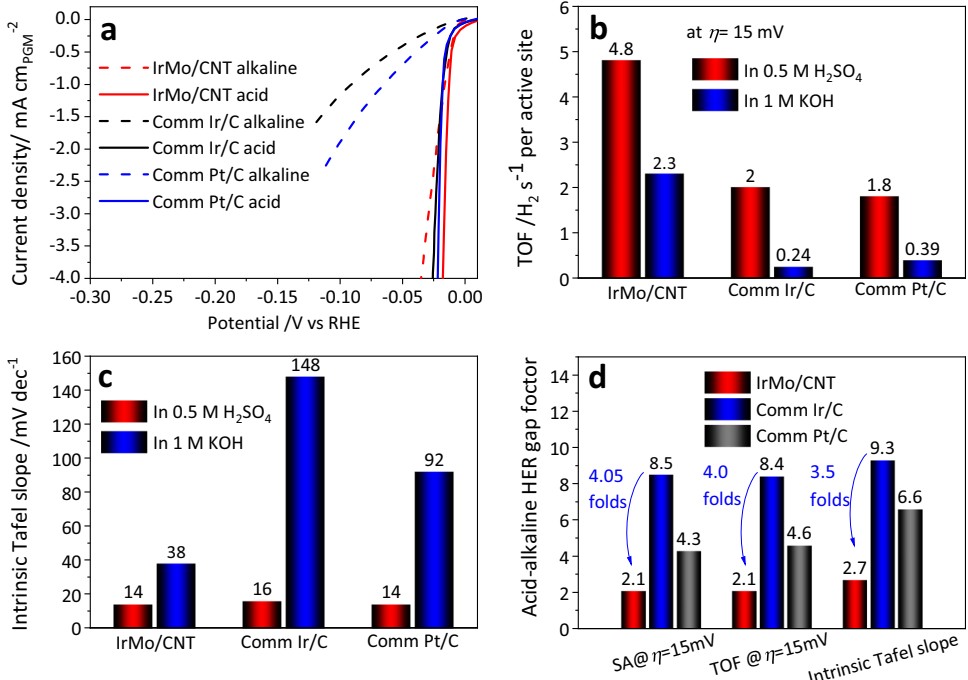

**Fig. 4 | Comparison of the kinetics and intrinsic activity between acidic and alkaline HER for IrMo/CNT, Ir/C, and Pt/C. a** Alkaline and acidic HER polarization curves with the current density normalized by ECSA of the measured IrMo/CNT, commercial Ir/C, and commercial Pt/C. **b** Comparison of the TOF at $\eta = 15$ mV. **c** Comparison of the intrinsic Tafel slopes extracted from the Tafel plots with the current density normalized by the ECSA of the electrocatalysts in acidic and alkaline electrolytes. **d** Comparison of the acid-alkaline HER gap factor on specific activity (SA), TOF, and Tafel slope. The acid-alkaline HER gap factor can be determined by SA$_{acid}$/SA$_{alkaline}$, TOF$_{acid}$/TOF$_{alkaline}$, or intrinsic Tafel slope$_{alkaline}$/intrinsic Tafel slope$_{acid}$.

with Ir/C and Pt/C in acid HER (Fig. 4c). The intrinsic HER Tafel slopes of IrMo/CNT in acid and base are 14 and 38 mV dec$^{-1}$, respectively; while those of Ir/C are 16 and 148 mV dec$^{-1}$ and those of Pt/C are 14 and 92 mV dec$^{-1}$, respectively. These results are quite different from the

Tafel curves plotted with the geometric current density in Supplementary Figs. 2 and 3.

The activity ratio (SA$_{acid}$/SA$_{alkaline}$, TOF$_{acid}$/TOF$_{alkaline}$) and Tafel slope ratio (intrinsic Tafel slope$_{alkaline}$/intrinsic Tafel slope$_{acid}$) have

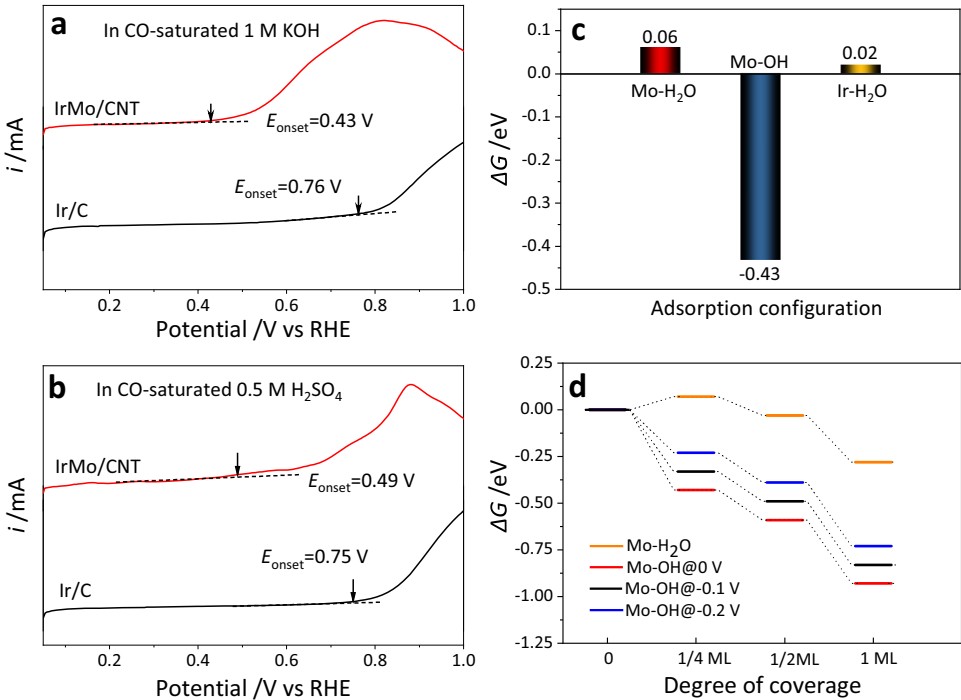

**Fig. 5 | Surface structure of IrMo model. a, b** LSV curves of IrMo/CNT and Ir/C in CO-saturated 1 M KOH (**a**) and 0.5 M $H_2SO_4$ (**b**) with a scan rate of 50 mV s$^{-1}$. The anode current in these curves represents the CO oxidization current. The positions marked by arrows indicate the onset potential of CO oxidization. **c** The adsorption free energies of $H_2O$ and OH at different sites of IrMo (001). The adsorbed OH at Ir site is quite unstable and thus the adsorption free energy is unavailable. **d** The adsorption free energies of OH and $H_2O$ on Mo site of IrMo (001) with different coverages under different applied potentials.

been employed to describe the acidic-alkaline HER gaps factor of different electrocatalysts (Fig. 4d). The closer the gap factor is to 1, the smaller the HER activity gap is. The gap factor determined by SA and TOF for IrMo/CNT both are 2.1. Differently, those gap factors determined by SA and TOF are 8.5 and 8.4 for Ir/C, and 4.3 and 4.6 for Pt/C, respectively. These gap factors of IrMo/CNT are more than 4 and 2 times smaller than those of the Ir/C and Pt/C, respectively. These indicate a narrower activity gap between acidic and alkaline HER for IrMo/CNT; while the Pt/C and Ir/C exhibit a larger acidic-alkaline HER activity gap. Similar results can be found in the gap factor described by Tafel slope as well.

### Mechanism of the narrowed activity gap between acidic HER and alkaline HER

To give insight into the favorable alkaline HER activity of IrMo electrocatalyst and reveal the mechanism of the narrowed kinetics gap between acidic HER and alkaline HER, we have performed a series of DFT calculations. The (001) surface is selected as the computational model of IrMo intermetallic, because it possesses the lowest surface energy (Supplementary Table 2). According to the reported works, the surface of the electrocatalyst may change after being immersed in electrocatalytic condition. For instance, spontaneous adsorption of OH at a dual-metal Fe, Co-NC electrocatalysts in the electrolyte was found to enable excellent oxygen reduction activity[20]. CO-stripping measurement has been employed to evaluate the OH adsorption ability for electrocatalysts, as the CO-oxidation process triggers with the reactive OH*[15,32]. By measuring the LSV curves in CO-saturated electrolyte (Fig. 5a, b), it has been found that the onset potential of CO oxidization on IrMo has negatively shifted ~330 and ~260 mV compared with those of Ir/C in 1 M KOH and 0.5 M $H_2SO_4$, respectively. These significant shifts caused by the alloying of Ir with Mo indicate the easier adsorption of OH on the surface of IrMo than Ir. The lower CO-oxidation barrier for IrMo may be ascribed to the synergy of the spontaneously adsorbed OH at Mo site and the adsorbed CO at Ir site.

Inspired by this, we have calculated the adsorption free energy of $H_2O$ and OH species on the (001) facet of IrMo intermetallic. It can be found that the adsorption free energies of $H_2O$ on both Mo and Ir sites are positive, which indicates an unspontaneous adsorption of $H_2O$ (Fig. 5c). Interestingly, the $\Delta G_{OH^*}$ on Mo site is calculated as −0.43 eV, while the adsorption of OH on Ir site is quite unstable. This implies that the alloying of Ir with Mo can render stronger adsorption of OH species, which is in line with the above CO-stripping measurements. Then, the $\Delta G_{OH^*}$ on Mo site of IrMo (001) surface with different OH* coverages under different applied potentials were also calculated (Fig. 5d). The monolayer of OH* on Mo site of IrMo (001) is the most stable configuration between the evaluated coverages. When the applied potential alters from 0 to −0.2 V vs RHE, the adsorption of OH becomes weaker but still stable. The stable adsorption of OH can also be confirmed by the FTIR spectra analysis. As shown in Supplementary Fig. 21a, the broad peak at 3000−3600 cm$^{-1}$ can be indexed to the stretching vibration of O−H[33]. In contrast to the spectra of the ex-situ IrMo/CNT and IrMo/CNT in KOH solid, the in-situ FTIR spectrum presents a much stronger O−H stretching peak, indicating the considerable adsorption behavior of hydroxides at Mo site. This result is consistent with the reported work, in which an OH ligand was stably adsorbed at Co site of the single-atom Co-N-C catalyst[34]. Given this, we can conclude that the adsorption of OH could spontaneously occur at the surface of IrMo under alkaline and acid HER condition. A slab of IrMo (001) with one monolayer OH adsorbed on the Mo site [IrMo (001)-OH] is thus used for further calculations (Supplementary Fig. 21b). The $\Delta G_{OH^*}$ at Ir (111) were also calculated (Supplementary Fig. 22), which indicates a large adsorption energy barrier of OH at these surfaces.

Before comparing the mechanism of the HER in base with that in acid, the mechanism of acidic HER for IrMo was firstly modeled. In acidic condition, the HER generally proceeds through Volmer−Tafel reaction for noble metal-based electrocatalysts[28]. The HER free energy diagram of Volmer−Tafel mechanism for Ir (111) and IrMo (001)-OH is

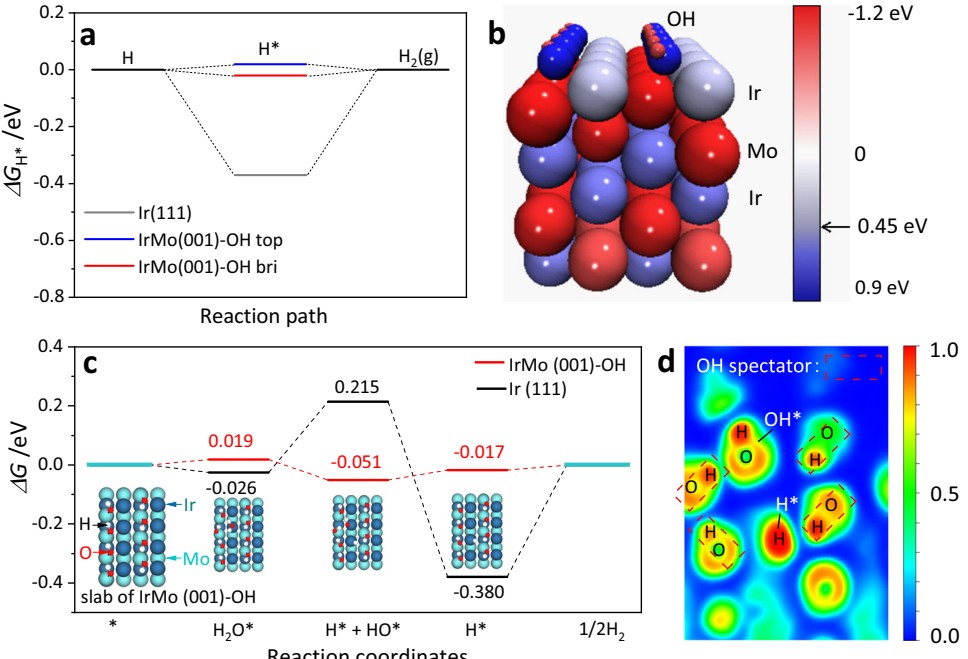

**Fig. 6 | HER mechanism analysis. a** Gibbs free energy diagram for HER reaction through Volmer–Tafel reaction. Slab of IrMo-OH exhibits two types of the stable adsorption configuration, including the top and bridge configuration at Ir. **b** Color-patched graph of Bader charge for IrMo (001)-OH slab, the arrow-marked position at the reference scale column indicates the captured electron number of the surface Ir atom. **c** Gibbs free energy diagram for HER reaction based on Volmer–Heyrovsky process. **d** Electron localization function (ELF) plot of the IrMo (001)-OH slab adsorbed with a H* and OH*.

depicted in Fig. 6a. Two types of the stable adsorption configuration for H* on Ir site of IrMo (001)-OH slab are considered, namely the top adsorption and bridge adsorption. The H* on top site and bridge site of Ir at IrMo (001)-OH show the near thermal-neutral $\Delta G_{H^*}$ of 0.02 and −0.02 eV, respectively. It is worth noting that the $\Delta G_{H^*}$ on these sites of IrMo are much more beneficial than that of the Ir (111) (−0.37 eV) according to Sabatier principle[4]. It can be found that the HER activity in acid electrolyte of the measured catalysts presents a good linear relationship with the calculated $\Delta G_{H^*}$, which matches well with the reported Sabatier volcano (Supplementary Note 2 and Supplementary Fig. 23a)[35]. The color-patched graph of the Bader charge for IrMo (001)-OH slab is provided in Fig. 6b to reveal the insight of such an improved effect, which reveals a number of 0.45 electron moved to the surface Ir site. This is consistent with the partial density of state calculation that the *d*-band center of Ir exhibits a negative shift of 0.15 eV from Ir (111) to IrMo (001)-OH (Supplementary Fig. 24), which weakens the interaction between the adsorbates and electrocatalysts and thus results in favorable H adsorption strength (Supplementary Note 2)[26].

According to the Tafel approximation in Supplementary Fig. 25, the IrMo and Ir electrocatalysts follow Volmer–Heyrovsky process during alkaline HER[13]. The difference in Gibbs free energy through the related pathway has thus been calculated and depicted in Fig. 6c. Due to the proton donor having changed to H₂O in alkaline electrolyte, the adsorption of H₂O is considered as the first step for HER[36]. As can be seen in Fig. 6c, the adsorption of H₂O on Ir (111) is slightly exothermal; while those of the IrMo (001)-OH is slightly endothermal. These indicate the adsorption of H₂O at Ir (111) is easier than the IrMo (001)-OH. The dissociation of the adsorbed H₂O (second step) has widely been considered an important step for alkaline HER process[13,36]. It is worth noting that such key step at IrMo (001)-OH is exothermal and has no thermodynamics barrier, which is highly favorable for water dissociation. In contrast, the Ir (111) exhibits an uphill $\Delta G$ at the second step, indicating the inhibited water dissociation. The third step and last step of IrMo (001)-OH, which can be indexed to the desorption of OH* and H*, are both endothermic by 0.034 and 0.017 eV, respectively. As for Ir

(111), the third step is thermodynamically favorable, but its desorption of H* must overcome a large energy barrier. Considering the reaction energy of each step, the inhibited step for IrMo (001)-OH can be the desorption of OH* with the largest endothermic energy of a mere 0.034 eV. Differently, the inhibited step for Ir (111) is the desorption of H*. The related thermodynamics barrier of the inhibited step for Ir (111) (0.380 eV) is much greater than that of the IrMo (001)-OH (0.034 eV), which leads to the superior alkaline HER activity of IrMo to Ir/C. This is consistent with the electrochemical measurement results in Figs. 3 and 4. The arrow-marked position in electron localization function plot of the IrMo (001)-OH after the water dissociation step exhibits an electronic localization between the dissociated OH at Ir site and the adsorbed OH spectator at Mo site from the initial model (Fig. 6d). This can be indexed to the formation of hydrogen bond between the H of the dissociated OH and the O of the adsorbed OH spectator at Mo site. Such a hydrogen bonding further stabilizes the dissociated OH at Ir site and thus facilitate the water dissociation step of IrMo intermetallic. Given the mechanism analysis of the acid and alkaline HER, it can be found that the IrMo (001)-OH in alkaline HER exhibits a close thermodynamics barrier to that of in acid HER, which has greatly narrowed the activity gap between the acid and alkaline HER. In addition, the stably adsorbed OH spectator at Mo site of IrMo intermetallic plays a key role for realizing the thermodynamically favorable water dissociation process and the superior activity in alkaline HER.

In summary, we demonstrate a highly efficient IrMo intermetallic electrocatalyst for alkaline HER with ultrasmall particle size of ~2 nm that was synthesized by a hydrogel-freeze-drying method with carbon nanotube as support. The as-synthesized IrMo/CNT catalyst exhibits superior catalytic performance to the commercial Ir/C and Pt/C. More importantly, the benchmarking activity of IrMo/CNT in alkaline media is fairly close to that in acidic electrolyte, which significantly narrows its intrinsic HER activity gap between base and acid. The extraordinarily high performance of IrMo/CNT catalyst is ascribed to the stably adsorbed OH spectator at Mo sites, which stabilizes the OH from water dissociation and thus facilitates the water dissociation. This

study highlights the essential role of the adsorbed OH spectator at the catalyst surface for promoting the alkaline HER and provides an effective strategy to rationally design advanced ordered intermetallic electrocatalysts for electrochemical energy devices including but not limited to water electrolyzer and fuel cells.

## Methods

### Chemicals

$IrCl_3 \cdot H_2O$ and $Na_2MoO_4 \cdot 2H_2O$ were purchased from Aladdin Ltd. CNT was purchased from Shanghai Hesen electric co., Ltd. Commercial Pt/C were purchased from JM Corporation. Commercial Ir/C was purchased from Premetek Company. KOH, $H_2SO_4$, $HClO_4$, and $HNO_3$ were purchased from Guangzhou Chemical Reagent Factory. All chemicals were used without further purification.

### Hydrophilic treatment of carbon nanotube (CNT)

The hydrophilic treatment of CNT was conducted by surface oxidization in nitric acid solution at 160 °C. Typically, a certain amount of the commercial CNT was added to 200 mL nitric acid solution (1.0 M) in a flask. The flask was then transferred to an oil bath at 160 °C under stirring and refluxed for 6 h. After cooling down to room temperature, the black suspension was filtrated and washed with deionized water 5-6 times. The CNT filter cake then underwent a freezing-dry treatment to obtain the hydrophilic CNT support.

### Synthesis of the CNT-supported intermetallic IrMo (IrMo/CNT)

IrMo intermetallic was synthesized by a two-step method of freeze-drying impregnation and annealing treatment using the activated CNT as support. In a typical process, 0.055 mmol $IrCl_3 \cdot H_2O$, 0.055 mmol $Na_2MoO_4 \cdot 2H_2O$ and 80 mg activated CNT were dispersed in 2 mL deionized water. The mixture then went through an ultrasound treatment for 1 h. After the mixture was formed into a hydrogel, it underwent freeze drying to remove the solvent and form an aerogel of metal precursors loaded CNT. Finally, the aerogel was annealed under $Ar/H_2$ (8 v%) flow at 800 °C for 6 h to obtain the CNT-supported IrMo intermetallic (IrMo/CNT). The Ir content in IrMo/CNT was determined to be 9.8 wt% by ICP-AES measurement.

### Characterization

XRD measurement was conducted on a MiniFlex 600 (Rigaku, Japan) using Cu K$\alpha$ radiation ($\lambda = 0.15406$ nm). HAADF/STEM images were obtained from an FEI TITAN transmission electron microscope (300 kV). TEM measurement was performed on a Jeol 2100F (200 kV). XPS results were obtained from a photoelectron spectrometer K-Alpha$^+$ (Thermo Fisher Scientific).

### Electrochemical tests

All electrochemical results were collected from an Autolab Electrochemical Instrumentation (Metrohm) with a conventional three-electrode system, in which a Pt mesh ($1 \times 1$ cm$^2$) was used as a counter electrode, a mercuric oxide electrode (Hg/HgO, in base) or silver chloride electrode (Ag/AgCl, in acid) was used as reference electrode, and rotating disk electrode (glassy carbon) was employed as the working electrode. All the electrochemical measurements were performed at room temperature and the electrochemical cell was used in an open environment. The catalyst inks were prepared by 1-h ultrasound treatment of a mixture including 5.0 mg catalysts, 50 μL Nafion/water/ethanol, and 950 μL ethanol. Then a certain amount of the homogeneous ink was dripped on the glassy carbon electrode with a Pt-group metal (PGM) loading of 25 μg cm$^{-2}$. The HER polarization curves were mainly collected in $H_2$-saturated 1.0 M KOH (between −0.9 and −1.2 V vs Hg/HgO) or 0.5 M $H_2SO_4$ (between −0.15 and −0.4 V vs Ag/AgCl) solution at a scan rate of 10 mV s$^{-1}$ and a rotation rate of 1600 rpm. The results were obtained with iR correction, which was carried out by the following equation: $E_{correction} = E - iR_{solution}$.

Accelerated durability test was performed by cycling the electrode between 0.05 and −0.05 V vs RHE at a scan rate of 100 mV s$^{-1}$ and a rotation of 1600 rpm for 10,000 cycles in $N_2$-saturated electrolytes. The ECSA of IrMo/CNT and commercial Ir/C were measured by the CO-stripping curves in 0.5 M $H_2SO_4$ according to the reported method[37]. Typically, the working electrode was held at 0.1 V vs RHE in CO-saturated 0.5 M $H_2SO_4$ for 10 min. Then the CO in the solution was removed by bubbling the electrolyte with $N_2$ flow for 15 min. The CO-stripping cyclic voltammetry (CV) was finally performed between a potential window of 0.05 and 1.3 V vs RHE with a scan rate of 50 mV s$^{-1}$. The ECSA of Pt/C was obtained by integrating the hydrogen desorption charge on CV curve in $N_2$-saturated 0.1 M $HClO_4$ according to the reported method[28]. The TOF was calculated by the following equation: $TOF = i/nFN$, where $i$ is the HER current under a certain overpotential, $n$ is the electron transfer number of the reaction ($n = 2$ for HER), $F$ is the Faraday constant, and $N$ is the mole number of the PGM metal as an active site at the surface of the electrocatalyst. $N$ can be calculated by the following equation: $N = ECSA \bullet M \bullet N_0 / N_A$, where $M$ is the mass loading of the PGM metal at the electrode, $N_0$ is the constant of PGM metal surface concentration ($1.25 \times 10^{19}$ m$^{-2}$ for Pt and $1.30 \times 10^{19}$ m$^{-2}$ for Ir), and $N_A$ is the Avogadro constant.

### DFT calculation

The DFT computations were performed using Vienna ab initio simulation package (VASP)[38]. Projector augmented wave and the generalized gradient approximation with the Perdew–Burke–Ernzerhof functional were used to describe the ion-electron interaction and the exchange-correlation potential[39,40]. Then, 400 eV cutoff was employed while the convergence threshold for the self-consistent field and ion steps was set to be $1 \times 10^{-5}$ eV and 0.01 eV Å$^{-1}$. Van der Waals interaction was considered by DFT-D3 method[41]. The dipolar correction was included and the symmetrization was switched off. The vacuum space of all investigated slab models was 15 Å. For the slab models of IrMo, different low-index surfaces with different termination and four layers were adopted. The model of Ir (111) contains six layers. The bottom half layers were fixed for all slabs. The Brillouin zones were sampled by Monkhorst-Pack $4 \times 3 \times 1$ and $3 \times 3 \times 1$ for the surface of IrMo and Ir(111)[42].

The surface energy ($\alpha$) was determined by using the following equation:

$$\alpha = \frac{E_{slab} - nE_{bulk}}{2A} \tag{1}$$

where $E_{slab}$ is the total energy of the slab, $E_{bulk}$ is the energy of a bulk unit, $n$ is the number of bulk unit in the slab, and $A$ is the surface area of the slab.

The adsorption Gibbs free energy ($\triangle G$) was calculated by the following equation:

$$\triangle G = \triangle E + \triangle ZPE - T \triangle S \tag{2}$$

where $\triangle E$ is the DFT-derived adsorption energy, which is the energy difference of configuration after and before adsorbed a reaction species. $\triangle ZPE$, $T$, and $\triangle S$ are the difference of the zero-point energy, temperature, and the difference of entropy.

The ZPE and S were calculated by using vibration frequencies with harmonic normal mode approximation, as in the following equations: $ZPE = 1/2 \sum_{i=N} hw_i$ and $S = k_B \sum_{i=N} \ln(1 - e^{-hw_i/k_B T})$, where $k_B$, $T$, $N$, $h$, and $w_i$ are Boltzman constant, temperature, the number of independent freedom degree, Planck's constant, and the vibration frequency, respectively[43]. Actually, the algorithm has been embedded in the VASPKIT[44], which is the post-processing program of VASP. After finishing the frequency calculation, we can obtain the ZPE and entropy correction with the 501 function in the VASPKIT.

## Data availability
The data that support the findings of this study are available within the article and its Supplementary Information files. All other relevant data supporting the findings of this study are available from the corresponding authors upon reasonable request. Source Data are provided with this paper.

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

## Acknowledgements

This work was supported by the National Natural Science Foundation of China (NSFC Project No. 22072048 to Z.C. and No. 21832004 to S.C.), the Guangdong Provincial Department of Science and Technology (Project No. 2021A1515010128 to Z.C.), the China Postdoctoral Science Foundation (Project No. 2022M711195 to J.Z.), and the Open Funds of the State Key Laboratory of Electroanalytical Chemistry (SKLEAC202208 to J.Z.).

## Author contributions

J.Z., Z.C., and S.C. conceived the idea and designed the experiments. J.Z. synthesized the sample, performed the electrochemical measurements, and primarily wrote the manuscript. J.L. and Y.T. participated in the synthesis of the materials, and the electrochemical measurements, respectively. J.Z., L.Z., and C.Z contributed to the physical measurements. L.Z. carried out the DFT simulation. J.Z., L.Z., and P.L. participated in the analysis of the DFT results. Z.C., L.D., and S.C. revised the manuscript. Z.C. and S.C. supervised the work. All the authors discussed the results and commented on the manuscript.

## Competing interests

The authors declare no competing interests.
