## [Peer Review File · Nature Communications]

OH Spectator at IrMo Intermetallic Narrowing Activity Gap between Alkaline and Acidic Hydrogen Evolution ReactionREVIEWER COMMENTS

Reviewer #1 (Remarks to the Author):

This paper investigates materials for the important challenge of HER in non acidic media. The authors report that a IrMo intermetallic is superior to Pt and to Ir commercial catalysts. Extensive electrochemical characterization is included alongside structural and theoretical justification for the performance. However, some of the broad and generalizable claims by the authors need further support and justification. I have several questions that I would need to see expanded on before I could recommend publication.

1. Can the performance of this be relationalized through d-band theory and Sabatier Volcanos? For example, the valence spectra in 1b. This is essential to link with theory if reporting that this is an underlying general principle for catalyst rational design.
2. The authors claim the mechanism is related to bound OH. Is there direct spectroscopic evidence to justify this mechanism? What about the specific active site?
3. Is the sample homogeneous throughout? Are there small clusters or regions of Ir or Mo rich material? The XRD is fine but bulk, more extensive electron microscopy with XAS would be important to prove the homogeneous nature of the catalysts.

Reviewer #2 (Remarks to the Author):

The manuscript presents interesting data on the HER in alkaline media on IrMo nanoparticles. However, the manuscript presents several important problems that prevent its publication in the present form. I will detail these problems:

1. Experimental details should be provided. Solutions compositions, annealing temperatures... This is especially important when it is stated that the procedure has been modified from previous reports.
2. Tafel slopes at low overpotentials, below 60 mV, are meaningless. In fact, theory predicts that current vs. overpotentials should be linear when the overpotential is lower than 20 mV. Only when the backward reaction can be neglected is the Tafel approximation valid. When the absolute value of the overpotential is below 60 mV, the backward reaction cannot be neglected and the Tafel approximation is no longer valid. Moreover, correct measurements under those conditions of reaction kinetics to determine the mechanism would require the presence of H₂ in the solution, so that the overpotential is correctly defined.
3. The observed differences between the sample in acidic solution are mostly due to the differences in the environment (carbon support, nanopores size...). Probably the reactivity is so high that the behavior is close to that expected for a thermodynamically controlled reaction.
4. After 1000 cycles, the Pt surfaces are heavily contaminated explaining the decay in the currents, as revealed by the voltammograms of the supporting information. This is the reason for the current decay for HER. If solutions were clean, no decay would have been observed. Probably the contamination present in solution/support affects to a much lesser extent the IrMo sample and the observed decay is much lower.
5. Authors have analyzed their results assuming that adsorbed OH catalyzes HER reaction in alkaline. However, some results indicate that this interpretation is not valid (see references 13 and 15 of the manuscript). The analysis of the data should be also taken into account these results.

Reviewer #3 (Remarks to the Author):

The paper titled "Stably Adsorbed OH Spectator at IrMo Intermetallic Narrowing the Activity Gap

between Alkaline and Acidic Hydrogen Evolution Reaction" by Zhang et al. reported a highly active IrMo/CNT catalyst for HER. They also performed DFT calculations to reveal the reaction mechanism of HER on IrMo catalyst. The manuscript was well written. However, this manuscript has several issues to be solved to be published in Nature Communications.

- (1) The authors well describe the motivation of this work. In this sense, why did the authors use CNT as a support material in this work? It should be addressed in the introduction of this paper.
- (2) Reference used in Fig. 2b should be specified in the caption of Fig. 2b.
- (3) What is the surface coverage of Ir and IrMo used in this paper?
- (4) How did the authors calculate entropy?
- (5) From XRD data in Fig. 1, I think (111) is a more dominant facet of IrMo in this experiment. Therefore, the catalytic activity of HER on IrMo(111) also should be investigated. The point of computer simulation in such collaborative works is to describe catalysis under experimental conditions, not an ideal condition.
- (6) How did the authors model IrMo(001)? It seems that the IrMo(001) is layer by layer structure of Ir and Mo. Why this atomic arrangement was adopted, not the homogeneous model?
- (7) What about the catalytic activity of pure IrMo? In order to assure the effect of OH species on enhancing the catalytic activity of IrMo, it is recommended.
- (8) Did you check the relationships between the d-band center and adsorption energies of reaction intermediates? If the d-band center is closely related to the adsorption strength of H, it also affects the other reaction intermediates.
- (9) In Fig. 5, it seems that most of the electrons transferred from Mo to oxygen atoms in adsorbed OH rather than surface Ir.
- (10) Why does the charge transfer toward the surface weaken the adsorption strength of H (or downshift the d-band center)?
- (11) In Fig. 5d, elements should be identified to understand explanations for this clearly.
- (12) Adsorption energies of all reaction intermediates should be provided in Supplementary Information

Point-by-point Response to the Reviewers

Reviewer #1

This paper investigates materials for the important challenge of HER in non acidic media. The authors report that a IrMo intermetallic is superior to Pt and to Ir commercial catalysts. Extensive electrochemical characterization is included alongside structural and theoretical justification for the performance. However, some of the broad and generalizable claims by the authors need further support and justification. I have several questions that I would need to see expanded on before I could recommend publication.

Q1. Can the performance of this be relationalized through d band theory and Sabatier Volcanos? For example, the valence spectra in 1b. This is essential to link with theory if reporting that this is an underlying general principle for catalyst rational design.

Response: Thanks for the good question. The performance of IrMo can be relationalized through d band theory and Sabatier Volcanos.

1) d band theory correlation. The signal of the valence spectra reflects the bulk information of the catalysts. The d band state at the surface may have better relationship with the activity. We thus use the calculated d band center from the DFT-modeled slab to relationalize with the H adsorption Gibbs free energy. As can be seen in Figure R1a below, the d band center of the Ir at pure IrMo (001) locates closer to the Fermi level than that of the Ir (111), leading to a stronger H adsorption. The modification of OH spectator at Mo site of the IrMo (001) would shift the Ir d band center to a more negative position than those of the pure IrMo (001) and Ir (111). It can be found in Figure R1a, a linear correlation is observed between the calculated ΔG_{H^*} and d band center, which can be explained by the d -band theory that the negative shift of the d band center would weaken the H adsorption ability (*Angew. Chem. Int. Ed.* **2019**, 58: 1 – 6).

2) Sabatier Volcanos correlation. Further, we relationalize the measured specific HER activity of the IrMo/CNT, Ir/C and Pt/C in acid and alkaline condition with the calculated ΔG_{H^*} . As shown in Figure R1b-c (equal to Figure S23a-b), a good linear correlation is observed between the HER activity and the calculated ΔG_{H^*} . According to the Sabatier Principle, a hydrogen adsorption that is neither too strong nor too weak is beneficial for facilitating the HER. The trends of the plots in Figure R1b-c match with the strong-adsorption side of the traditional HER Sabatier volcano plot (*J.*

Electrochem. Soc. **2005**, 152: 23-26).

The related results and discussion are presented in supplementary Figure S23-24 and Note 2 of the revised supplementary information (page S14-15) and mentioned in the revised manuscript (Page 8, the second paragraph).

Figure R1. (a) Plot of ΔG_{H^*} versus d band center for pure IrMo (001), IrMo (001)-OH and Ir (111) slab. (b-c) Plot of hydrogen evolution reaction activity (b) in acid and (c) alkaline condition at $\eta=20$ mV versus the ΔG_{H^*} .

Q2. The authors claim the mechanism is related to bound OH. Is there direct spectroscopic evidence to justify this mechanism? What about the specific active site?

Response: Thanks for the comments. To justify the mechanism, we have carried out the ex-situ and in-situ FTIR analysis, and the results are shown in Figure R2a and Figure S21a in the revised supplementary information. The broad peak at 3000-3600 cm^{-1} in Figure R2a can be indexed to the stretching vibration of O-H (*ACS Nano*, **2014**, 8: 6856–6862). In contrast to the spectra of the ex-situ IrMo/CNT and IrMo/CNT in KOH solid, the in-situ FTIR spectrum under open-circuit condition presents a much stronger O-H stretching peak, indicating the considerable adsorption behavior of hydroxides at Mo site. This result is consistent with the reported work, in which a OH ligand stably adsorbs at Co site of the single-atom Co-N-C catalyst (*Nat. Catal.* **2018**, 2: 134-141). The related discussion has been added in the revised manuscript (Page 8, first paragraph).

Based on the FTIR spectra and DFT simulation, we deduce that the exposed plane under the in-situ condition is the IrMo (001) facet with the OH spectator stably adsorbed at Mo site (Figure R2b). Therefore, the specific active site could be the Ir site near the Mo atom.

Figure R2. (a) FTIR spectra of ex-situ IrMo/CNT, IrMo/CNT in KOH solid and IrMo/CNT in 1.0 M KOH solution under open-circuit condition; (b) The assumed active surface of IrMo (001)-OH for IrMo intermetallic.

Q3. is the sample homogeneous throughout? Are there small clusters or regions of Ir or Mo rich material? The XRD is fine but bulk, more extensive electron microscopy with XAS would be important to prove the homogeneous nature of the catalysts.

Response: The elemental distribution of IrMo intermetallic NPs is characterized by the HADDF/STEM line scan profile. As show in Figure 1f, the IrMo sample exhibit homogeneous elemental distribution throughout the whole particle and don't have an Ir/Mo-rich surface.

Reviewer #2

The manuscript presents interest data on the HER in alkaline media on IrMo nanoparticles. However, the manuscript presents several important problems that prevent its publication in the present form. I will detail these problems:

Q1. Experimental details should be provided. Solutions compositions, annealing temperatures... This is especially important when it is stated that the procedure has been modified from previous reports.

Response: Thanks for the suggestion. The experimental details have been provided in Methods section in the revised manuscript (Page 10-11).

Q2. Tafel slopes at low overpotentials, below 60 mV, are meaningless. In fact, theory predicts that current vs. overpotentials should be linear when the overpotential is lower than 20 mV. Only when

the backward reaction can be neglected is the Tafel approximation valid. When the absolute value of the overpotential is below 60 mV, the backward reaction cannot be neglected and the Tafel approximation is no longer valid. Moreover, correct measurements under those conditions of reaction kinetics to determine the mechanism would require the presence of H₂ in the solution, so that the overpotential is correctly defined.

Response: Thank you for the comments and suggestion. The intrinsic Tafel plots used for Tafel approximation is depicted in Figure S25 of the revised supplementary information, in which the current is plotted with the overpotential greater than 60 mV. Despite this change, the Tafel approximation results are consistent with that of the original manuscript.

In addition, we agree that the HER measurements must be carried out in the H₂-saturated electrolyte to avoid shifting the equilibrium potential. Indeed, we did use the H₂-saturated 1 M KOH and 0.5 M H₂SO₄ to measure the HER polarization curves. The experimental details had been presented in the Methods section (Page 10, *Electrochemical Tests*).

Q3. The observed differences between the sample in acidic solution are mostly due to the differences in the environment (carbon support, nanopores size....). Probably the reactivity is so high that the behavior is close to that expected for a thermodynamically controlled reaction.

Response: We agree that the differences in the environment may impact the performance of the electrocatalysts. To evaluate the effect of such a difference on the HER activity, we further compare the performance of the commercial Pt/C with the home-made Pt/CNT. As can be seen in Figure R3, the Pt/CNT shows a similar onset HER potential but slightly lower specific activity compared to that of Pt/C, indicating that CNT support does not provide a better environment than the commercial carbon support for HER in acid solution. Therefore, relative to the Pt/C and Ir/C, the enhanced HER performance of IrMo/CNT arises from IrMo intermetallic catalyst rather than the CNT support. The OH spectator at Mo site optimizes the electronic states of the surface Ir site, which can be supported by the DFT analysis (the supplementary Figure S23-24). Therefore, as shown in the HER LSV curves of Figure 2d, the IrMo/CNT exhibits the lowest onset potential among the measured catalysts.

Figure R3. (a) TEM and XRD (the inserted pattern) characterization of the home-made Pt/CNT prepared by the similar method of IrMo/CNT (reduced at 200 °C for 2 h). (b) Comparison in specific HER activity of the Pt NPs on different carbon supports in H₂-saturated 0.5 M H₂SO₄.

Q4. After 1000 cycles, the Pt surfaces are heavily contaminated explaining the decay in the currents, as revealed by the voltammograms of the supporting information. This is the reason for the current decay for HER. If solutions were clean, no decay would have been observed. Probably the contamination present in solution/support affects to a much lesser extent the IrMo sample and the observed decay is much lower.

Response: Thanks for the comments. We performed the accelerated durability test (ADT) of both IrMo/CNT and Pt/C for 10,000 cycles. To exclude the factor of contamination, we first performed the accelerated durability test of 10,000 cycles, subsequently replaced the contaminated solution with new electrolyte, and obtained the HER polarization and CV curves. As can be seen in Figure R4a, the HER current of the Pt/C still present a remarkable decline after 10,000 cycles. The result clearly shows that the activity degradation may arise from other factors rather than contamination. The CV curves in Figure R4b exhibit a remarkably weakened H_{upd} peak and thus decreased ECSA of Pt/C after ADT. The TEM images in Figure R4c-d show that the Pt/C undergoes a heavy agglomeration of the Pt NPs during 10,000-cycle ADT test, which accounts for the deterioration of the HER performance.

We have modified the related discussion in the supplementary Note 1 (**Page S3**) according to the above analysis. Figure R4a and Figure R4b-d are also presented in Figure S5 (**Page S4**) and Figure S8 (**Page S6**) of the revised supplementary information, respectively.

Figure R4. (a) HER curves and (b) CV curves of Pt/C before and after 10,000 cycles test (electrolyte refreshed). TEM images of Pt/C (c) before and (d) after 10,000 cycles measurement.

Q5. Authors have analyzed their results assuming that adsorbed OH catalyzes HER reaction in alkaline. However, some results indicate that this interpretation is not valid (see references 13 and 15 of the manuscript). The analysis of the data should be also taken into account these results.

Response: Thanks for this comment, which involves an important issue that what impacts the alkaline HER kinetics of an electrocatalyst. To date, there are many interpretations on the kinetics of the alkaline HER on the various types of catalysts. For instance, Koper et al proposed a pH-dependent HER rate descriptor of interfacial water reorganization that can affect the HER (*reference 15 of the original manuscript*). On the other hand, they also reported the key role of the adsorbed OH intermediates in improving the alkaline HER kinetics (*reference 13 of the original manuscript*). In addition, the interfacial water structure (*ACS Appl. Mater. Interfaces*, **2019**, 11: 613–623) and electric field (*J. Am. Chem. Soc.* **2019**, 141: 15524–15531) are also employed to explain the alkaline HER at some modeled electrodes.

This work highlights the essential role of the adsorbed OH spectator for the alkaline HER. The OH spectator at Mo site optimizes the electronic states of the surface Ir site and stabilizes the water dissociation product, resulting in a thermodynamically favorable water dissociation process. Such promoting role of OH spectator is also observed in other electrocatalytic reaction. For example,

Ge et al. discovered that the stably adsorbed OH spectator at Fe-Co dual-atom electrocatalyst can facilitate the oxygen reduction reaction due to the aroused enhancement in electronic and geometric effect (*J. Am. Chem. Soc.* **2019**, 141: 17763–17770). Our analysis is based on the change in the electronic state of active sites and the chemical environment caused by the stably-adsorbed OH, which is inspired by the above-mentioned work. This study provides new understanding of alkaline HER mechanism in the community of the electrocatalysis.

Reviewer #3

The paper titled "Stably Adsorbed OH Spectator at IrMo Intermetallic Narrowing the Activity Gap between Alkaline and Acidic Hydrogen Evolution Reaction" by Zhang et al. reported a highly active IrMo/CNT catalyst for HER. They also performed DFT calculations to reveal the reaction mechanism of HER on IrMo catalyst. The manuscript was well written. However, this manuscript has several issues to be solved to be published in Nature Communications.

Q1. The authors well describe the motivation of this work. In this sense, why did the authors use CNT as a support material in this work? It should be addressed in the introduction of this paper.

Response: The essence of the work is a fundamental research about catalytic capability and mechanism of IrMo Intermetallic for HER. CNT just act as a support which promotes the controllable synthesis of ultrasmall IrMo Intermetallic nanoparticles. Therefore, we didn't address this in the introduction of this paper; however, we have discussed the role of the activated CNT in catalyst synthesis in the caption of Scheme 1 (Page 3). Compared to the conventional carbon support, the hydrophilic CNT is easier to crosslink and form hydrogel when it goes through ultrasound treatment in aqueous solution with suitable concentration. This is beneficial for forming aerogel and homogeneously dispersing the metal salts at the support after freeze drying.

Q2. Reference used in Fig. 2b should be specified in the caption of Fig. 2b.

Response: Thank you for the suggestion. We have added the related references in the caption of Fig. 2b.

Q3. What is the surface coverage of Ir and IrMo used in this paper?

Response: In this work, the surface coverage of IrMo (001) is 1 monolayer (ML), that is, one Mo

atom corresponds to one OH*. To determine the coverage of IrMo (001), we calculated the change of Gibbs free energy (ΔG) with the change of OH* coverage, which includes 1/4 ML, 1/2 ML and 1 ML. Fig. 4d of the manuscript shows the results that the ΔG at 1 ML of coverage is always most negative. This means that 1 ML of OH* located on Mo site of IrMo (001) is the most stable configuration. As for Ir, the surface coverage in this work is 0. The calculated ΔG_{OH^*} on Ir (111) is positive (0.77 eV, Supplementary Figure 22), which indicates the adsorption of OH on Ir (111) is unstable. Therefore, the OH surface coverage of Ir and IrMo used in this paper is 0 and 1 ML, respectively.

Q4. How did the authors calculate entropy?

Response: The entropy was calculated by using vibration frequencies with harmonic normal mode approximation, as in the equation $S = k_B \sum_{i=N} \ln(1 - e^{-hw_i/k_B T})$, where k_B , T , N , h , and w_i are Boltzman constant, temperature, the number of independent freedom degree, Planck's constant and the vibration frequency, respectively (*ACS Appl. Mater. Interfaces*, **2017**, 9: 42688-42698.). Actually, the algorithm has been embedded in the VASPKIT (*Comput. Phys. Commun.* **2021**, 267: 108033.), which is a post-processing program of VASP. After finishing the frequency calculation, we can obtain the entropy correction with the 501 function in the VASPKIT. We only consider the entropy of absorbents and gas molecular because the entropy difference of slab is so small that we can neglect it (*J. Electrochem. Soc.* **2005**, 152: J23.).

The above calculation details have been added in the Method section of the revised manuscript (Page 12, *first paragraph*).

Q5. From XRD data in Fig. 1, I think (111) is a more dominant facet of IrMo in this experiment. Therefore, the catalytic activity of HER on IrMo(111) also should be investigated. The point of computer simulation in such collaborative works is to describe catalysis under experimental conditions, not an ideal condition.

Response: Thanks for the suggestion. In general, the crystal surfaces with high surface energy will decrease and the crystal surfaces with low surface energy will increase during annealing process in order to be more thermodynamically stable. In this work, the preparation of the IrMo/CNT

involved an annealing process at 800 °C for 6 hours. To determine the dominant facet of IrMo, we calculated the surface energy of different surfaces and the results are presented in Supplementary Table 2. The (001) surface with the lowest surface energy is considered to have the largest proportion of the exposed surfaces, and thus it is used as the catalytic active crystal plane. This approach was also been endorsed and adopted by some other works (*Angew. Chem. Int. Ed.* **2019**, 131: 11531-11535.; *Isience*, **2019**, 17: 315-324.).

In addition, the XRD diffraction intensity is determined by the type, number and arrangement of atoms in the crystal cell (*Woolfson M M, Woolfson M M. An introduction to X-ray crystallography. Cambridge University Press, 1997.*). The diffraction intensity of the whole crystal in all directions is actually several times more than that of a crystal cell. That is to say, the surface composition of the nanoparticles cannot be precisely determined by the intensity of the X-ray diffraction peak.

Q6. How did the authors model IrMo(001)? It seems that the IrMo(001) is layer by layer structure of Ir and Mo. Why this atomic arrangement was adopted, not the homogeneous model?

Response: The IrMo (001) was derived from IrMo crystal cell (see Figure R5). The structure file (.cif file) can be obtained from an open database called Materials Project (<https://materialsproject.org/>). The arrangement of the atoms in the crystal cell shows that the IrMo (001) is a layer by layer structure of Ir and Mo. Besides, the termination atom was determined by the surface energy calculation and the results can be found in Supplementary Table 2. The surface energy of IrMo (001) terminated with Ir is lower than that of IrMo (001) terminated with Mo. So, the IrMo (001) is a layer by layer structure and terminated with Ir.

Figure R5. Crystal cell of IrMo intermetallic.

Q7. What about the catalytic activity of pure IrMo? In order to assure the effect of OH species on enhancing the catalytic activity of IrMo, it is recommended.

Response: Thank you for the suggestion. Actually, the adsorption of OH spectator at surface of IrMo is spontaneous, which can be supported by the in-situ FTIR analysis under open-circuit condition (see Figure S21a in the revised supplementary information) and DFT calculation (Figure 4d) results. It is very challenging to avoid such a spontaneous adsorption behavior in the electrolyte when we intend to measure the HER activity of the pure IrMo. Therefore, it is difficult to provide the HER activity of the pure IrMo experimentally. Nevertheless, we have calculated the H adsorption Gibbs free energy (ΔG_{H^*}) of the pure IrMo (001) to predict the HER activity. As shown in Figure S23b of the revised supplementary information, the ΔG_{H^*} of pure IrMo is -0.50 eV, which is much more negative than that of the IrMo (001) with OH spectator (-0.02 eV). Such a strong H binding strength is unfavorable for HER due to the difficulty of releasing H₂.

Q8. Did you check the relationships between the d-band center and adsorption energies of reaction intermediates? If the d-band center is closely related to the adsorption strength of H, it also affects the other reaction intermediates.

Response: Thank you for the comment. We have checked the relationship between the *d*-band center and H adsorption energies. As can be seen in Figure R6a and Figure S23b in the revised supplementary information, the *d* band center of the Ir at pure IrMo (001) locates closer to the Fermi level than that of the Ir (111), leading to a stronger H adsorption. The modification of OH spectator at Mo site of the IrMo (001) would shift the Ir *d* band center to a more negative position than those of the pure IrMo (001) and Ir (111). It can be found in Figure R6a, a linear correlation is observed between the calculated ΔG_{H^*} and *d* band center, which can be explained by the *d*-band theory that the negative shift of the *d* band center would weaken the H adsorption ability (*Angew. Chem. Int. Ed.* **2019**, 58: 1-6).

We agree that generally d-band center affects the other reaction intermediates (e.g. OH) if it is closely related to the adsorption strength of H. However, our DFT calculation results suggest that the OH adsorption energy is not solely influenced by d-band center in IrMo intermetallic. In addition, the binding strength of OH intermediates at IrMo intermetallic would also be affected by the OH spectator at Mo site via H-bond interaction (Figure R6b, equal to Fig. 5d in the main text).

Therefore, there is no evident linear relationship between the d-band center and adsorption energy of OH intermediates for the investigated models.

Figure R6. (a) Plot of ΔG_{H^*} versus d band center for pure IrMo (001), IrMo (001)-OH and Ir (111) slab. (b) Electro localization function (ELF) plot of the IrMo (001)-OH slab adsorbed with a H^* and OH^* . **The black arrows mark the H-bond interaction between the OH intermediate and OH spectator.**

Q9. In Fig. 5, it seems that most of the electrons transferred from Mo to oxygen atoms in adsorbed OH rather than surface Ir.

Response: It's true that most electrons were transferred from Mo to O, and only a smaller number of electrons were transferred from Mo to Ir because Ir is more electronegative than Mo.

Q10. Why does the charge transfer toward the surface weaken the adsorption strength of H (or downshift the d-band center)?

Response: Response: As shown in Figure R7, when Ir or Pt get electrons from metal ligand, the d band center is negatively shifted relative to the Fermi energy level because the relative position of the Fermi energy level is unchanged (*Phys. Chem. Chem. Phys.* 2014, 16: 20360-20376.).

Figure R7. The illustration of the position of the d-band center (ϵ_d) when Ir or Pt get electrons from their ligand.

Q11. In Fig. 5d, elements should be identified to understand explanations for this clearly.

Response: Thanks for the suggestion. We have marked the position of the elements in Fig. 5d in the revised version. The Figure R8 presented below is equal to the revised Fig. 5d.

Figure R8. Electro localization function (ELF) plot of the IrMo (001)-OH slab adsorbed with a H* and OH*.

Q12. Adsorption energies of all reaction intermediates should be provided in Supplementary Information

Response: The adsorption energies of all reaction intermediates have been listed in Table S3 in the revised version of the supplementary information. The Table R1 presented below is equal to the Table S3.

Table R1. The calculated ΔG of the reaction intermediates for alkaline HER

intermediates\slab	IrMo (001)-OH	Ir (111)
H ₂ O*	0.019 eV	-0.026
H* + HO*	-0.051	0.215
H*	-0.017	-0.380

REVIEWERS' COMMENTS

Reviewer #1 (Remarks to the Author):

these additions and reviews now make this manuscript publishable in my opinion.

Reviewer #2 (Remarks to the Author):

The previous issues have been already solved. The manuscript can be published.

Reviewer #3 (Remarks to the Author):

Thank you for your effort. Now, most of my questions have been solved. However, I recommend only one more thing that add top and side views of DFT models. Then, this paper will be published without further revision.

Thanks.

Point-by-point response to reviewers

REVIEWERS' COMMENTS

Reviewer #1 (Remarks to the Author):

These additions and reviews now make this manuscript publishable in my opinion.

Response: Thanks for the positive comments.

Reviewer #2 (Remarks to the Author):

The previous issues have been already solved. The manuscript can be published.

Response: Thanks for the positive comments.

Reviewer #3 (Remarks to the Author):

Thank you for your effort. Now, most of my questions have been solved. However, I recommend only one more thing that add top and side views of DFT models. Then, this paper will be published without further revision.

Response: Thanks for the positive comments and the suggestion. The top and side views of the DFT model are presented in Figure S21b of the final supplementary information.